

# Chasing Ghosts: Instruction Following as Bayesian State Tracking

**Peter Anderson**[*1]     **Ayush Shrivastava**[*1]     **Devi Parikh**[1,2]     **Dhruv Batra**[1,2]     **Stefan Lee**[1,3]

[1]Georgia Institute of Technology, [2]Facebook AI Research, [3]Oregon State University

{peter.anderson, ayshrv, parikh, dbatra}@gatech.edu     leestef@oregonstate.edu

## Abstract

A visually-grounded navigation instruction can be interpreted as a sequence of expected observations and actions an agent following the correct trajectory would encounter and perform. Based on this intuition, we formulate the problem of finding the goal location in Vision-and-Language Navigation (VLN) [1] within the framework of Bayesian state tracking – learning observation and motion models conditioned on these expectable events. Together with a mapper that constructs a semantic spatial map on-the-fly during navigation, we formulate an end-to-end differentiable Bayes filter and train it to identify the goal by predicting the most likely trajectory through the map according to the instructions. The resulting navigation policy constitutes a new approach to instruction following that explicitly models a probability distribution over states, encoding strong geometric and algorithmic priors while enabling greater explainability. Our experiments show that our approach outperforms a strong LingUNet [2] baseline when predicting the goal location on the map. On the full VLN task, i.e., navigating to the goal location, our approach achieves promising results with less reliance on navigation constraints.

## 1   Introduction

One long-term challenge in AI is to build agents that can navigate complex 3D environments from natural language instructions. In the Vision-and-Language Navigation (VLN) instantiation of this task [1], an agent is placed in a photo-realistic reconstruction of an indoor environment and given a natural language navigation instruction, similar to the example in Figure 1. The agent must interpret this instruction and execute a sequence of actions to navigate efficiently from its starting point to the corresponding goal. This task is challenging for existing models [3–9], particularly as the test environments are unseen during training and no prior exploration is permitted in the hardest setting.

To be successful, agents must learn to ground language instructions to both visual observations and actions. Since the environment is only partially-observable, this in turn requires the agent to relate instructions, visual observations and actions through memory. Current approaches to the VLN task use unstructured general purpose memory representations implemented with recurrent neural network (RNN) hidden state vectors [1, 3–9]. However, these approaches lack geometric priors and contain no mechanism for reasoning about the likelihood of alternative trajectories – a crucial skill for the task, e.g., 'Would this look more like the goal if I was on the other side of the room?'. Due to this limitation, many previous works have resorted to performing inefficient first-person search through the environment using search algorithms such as beam search [5, 7]. While this greatly improves performance, it is clearly inconsistent with practical applications like robotics since the resulting agent trajectories are enormously long – in the range of hundreds or thousands of meters.

To address these limitations, it is essential to move towards reasoning about alternative trajectories in a *representation* of the environment – where there are no search costs associated with moving a physical robot – rather than in the environment itself. Towards this, we extend the Matterport3D simulator [1] to provide depth outputs, enabling us to investigate the use of a *semantic spatial map* [10–13] in the context of the VLN task for the first time. We propose an instruction-following agent incorporating three components: (1) a *mapper* that builds a semantic spatial map of its environment from first-

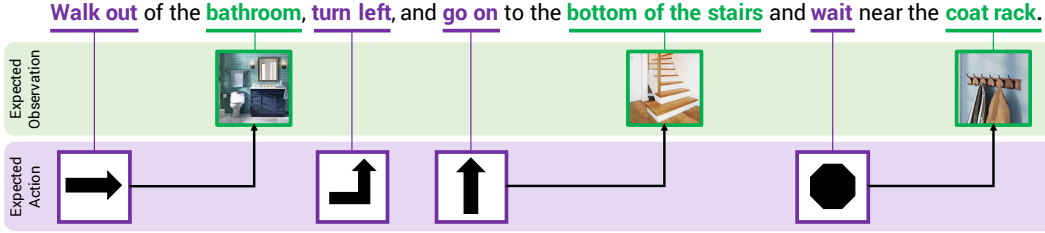

Figure 1: Navigation instructions can be interpreted as encoding a set of latent expectable observations and actions an agent would encounter and undertake while successfully following the directions.

person views; (2) a *filter* that determines the most probable trajectory(ies) and goal location(s) in the map, and (3) a *policy* that executes a sequence of actions to reach the predicted goal.

From a modeling perspective, our key contribution is the filter that formulates instruction following as a problem of Bayesian state tracking [14]. We notice that a visually-grounded navigation instruction typically contains a description of expected future observations and actions on the path to the goal. For example, consider the instruction 'walk out of the bathroom, turn left, and go on to the bottom of the stairs and wait near the coat rack' shown in Figure 1. When following this instruction, we would expect to immediately observe a bathroom, and at the end a coat rack near a stairwell. Further, in reaching the goal we can anticipate performing certain actions, such as turning left and continuing that way. Based on this intuition, we use a sequence-to-sequence model with attention to extract sequences of latent vectors representing observations and actions from a natural language instruction.

Faced with a known starting state, a (partially-observed) semantic spatial map generated by the mapper, and a sequence of (latent) observations and actions, we now quite naturally interpret our instruction following task within the framework of Bayesian state tracking. Specifically, we formulate an end-to-end differentiable histogram filter [15] with learnable observation and motion models, and we train it to predict the most likely trajectory taken by a human demonstrator. We emphasize that we are not tracking the state of the *actual agent*. In the VLN setting, the pose of the agent is known with certainty at all times. The key challenge lies in determining the location of the natural-language-specified goal state. Leveraging the machinery of Bayesian state estimation allows us to reason in a principled fashion about what a (hallucinated) human demonstrator would do when following this instruction – by explicitly modeling the demonstrator's trajectory over multiple time steps in terms of a probability distribution over map cells. The resulting model encodes both strong geometric priors (e.g., pinhole camera projection) and strong algorithmic priors (e.g., explicit handling of uncertainty, which can be multi-modal), while enabling explainability of the learned model. For example, we can separately examine the motion model, the observation model, and their interaction during filtering.

Empirically, we show that our filter-based approach significantly outperforms a strong LingUNet [2] baseline when tasked with predicting the goal location in VLN given a partially-observed semantic spatial map. On the full VLN task (incorporating the learned policy as well), our approach achieves a success rate on the test server [1] of 32.7% (29.9% SPL [16]), a credible result for a new class of model trained exclusively with imitation learning and without data augmentation. Although our policy network is specific to the Matterport3D simulator environment, the rest of our pipeline is general and operates without knowledge of the simulator's navigation graph (which has been heavily utilized in previous work [1, 3–9]). We anticipate this could be an advantage for sim-to-real transfer (i.e., in real robot scenarios where a navigation graph is not provided, and could be non-trivial to generate).

**Contributions.** In summary, we:

- Extend the existing Matterport3D simulator [1] used for VLN to support depth image outputs.
- Implement and investigate a semantic spatial memory in the context of VLN for the first time.
- Propose a novel formulation of instruction following / goal prediction as Bayesian state tracking of a hypothetical human demonstrator.
- Show that our approach outperforms a strong baseline for goal location prediction.
- Demonstrate credible results on the full VLN task with the addition of a simple reactive policy, with less reliance on navigation constraints than prior work.

## 2 Related work

**Vision-and-Language Navigation Task.** The VLN task [1], based on the Matterport3D dataset [17], builds on a rich history of prior work on situated instruction-following tasks beginning with SHRDLU [18]. Despite the task's difficulty, a recent flurry of work has seen significant improvements in success rates and related metrics [3–9]. Key developments include the use of instruction-generation ('speaker') models for trajectory re-ranking and data augmentation [7, 8], which have been widely adopted. Other work has focused on developing modules for estimating progress towards the goal [5] and learning when to backtrack [6, 9]. However, comparatively little attention has been paid to the memory architecture of the agent. LSTM [19] memory has been used in all previous work.

**Memory architectures for navigation agents.** Beyond the VLN task, various categories of memory structures for deep neural navigation agents can be identified in the literature, including unstructured, addressable, metric and topological. General purpose *unstructured* memory representations, such as LSTM memory [19], have been used extensively in both 2D and 3D environments [20–24]. However, LSTM memory does not offer context-dependent storage or retrieval, and so does not naturally facilitate local reasoning when navigating large or complex environments [25]. To overcome these limitations, both *addressable* [25, 26] and *topological* [27] memory representations have been proposed for navigating in mazes and for predicting free space. However, in this work we elect to use a *metric* semantic spatial map [10–13] – which preserves the geometry of the environment – as our agent's memory representation since reasoning about observed phenomena from alternative viewpoints is an important aspect of the VLN task. Semantic spatial maps are grid-based representations containing convolutional neural network (CNN) features which have been recently proposed in the context of visual navigation [10], interactive question answering [13], and localization [12]. However, there has been little work on incorporating these memory representations into tasks involving natural language. The closest work to ours is Blukis et al. [11], however our map construction is more sophisticated as we use depth images and do not assume that all pixels lie on the ground plane. Furthermore, our major contribution is formulating instruction-following as Bayesian state tracking.

## 3 Preliminaries: Bayes filters

A Bayes filter [14] is a framework for estimating a probability distribution over a latent state $s$ (e.g., the pose of a robot) given a history of observations $o$ and actions $a$ (e.g., camera observations, odometry, etc.). At each time step $t$ the algorithm computes a posterior probability distribution $bel(s_t) = p(s_t \mid a_{1:t}, o_{1:t})$ conditioned on the available data. This is also called the *belief*.

Taking as a key assumption the Markov property of states, and conditional independence between observations and actions given the state, the belief $bel(s_t)$ can be recursively updated from $bel(s_{t-1})$ using two alternating steps to efficiently combine the available evidence. These steps may be referred to as the *prediction* based on action $a_t$ and the *observation update* using observation $o_t$.

**Prediction.** In the prediction step, the filter processes the action $a_t$ using a *motion model* $p(s_t \mid s_{t-1}, a_t)$ that defines the probability of a state $s_t$ given the previous state $s_{t-1}$ and an action $a_t$. In particular, the updated belief $\overline{bel}(s_t)$ is obtained by integrating (summing) over all prior states $s_{t-1}$ from which action $a_t$ could have lead to $s_t$, as follows:

$$\overline{bel}(s_t) = \int p(s_t \mid s_{t-1}, a_t)\, bel(s_{t-1})\, ds_{t-1} \tag{1}$$

**Observation update.** During the observation update, the filter incorporates information from the observation $o_t$ using an *observation model* $p(o_t \mid s_t)$ which defines the likelihood of an observation $o_t$ given a state $s_t$. The observation update is given by:

$$bel(s_t) = \eta\, p(o_t \mid s_t)\, \overline{bel}(s_t) \tag{2}$$

where $\eta$ is a normalization constant and Equation 2 is derived from Bayes rule.

**Differentiable implementations.** To apply Bayes filters in practice, a major challenge is to construct accurate probabilistic motion and observation models for a given choice of belief representation $bel(s_t)$. However, recent work has demonstrated that Bayes filter implementations – including Kalman filters [28], histogram filters [15] and particle filters [29, 30] – can be embedded into deep neural networks. The resulting models may be seen as new recurrent architectures that encode algorithmic priors from Bayes filters (e.g., explicit representations of uncertainty, conditionally independent observation and motion models) yet are fully differentiable and end-to-end learnable.

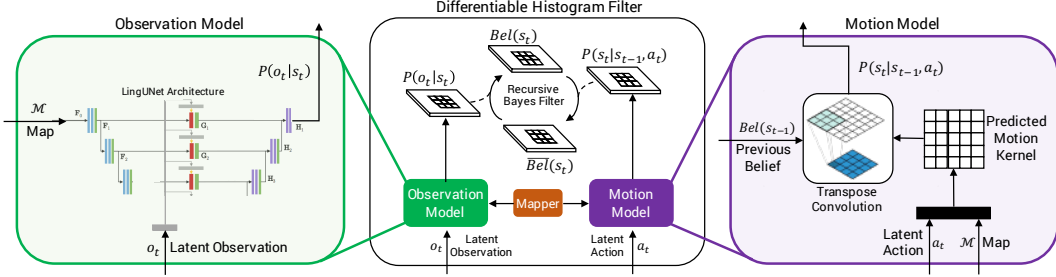

Figure 2: Proposed filter architecture. To identify likely goal locations in the partially-observed semantic spatial map $\mathcal{M}$ generated by the mapper, we first initialize the belief $bel(\boldsymbol{s}_t)$ with the known starting state $\boldsymbol{s}_0$. We then recursively: (1) generate a latent observation $\boldsymbol{o}_t$ and action $\boldsymbol{a}_t$ from the instruction, (2) compute the prediction step using the motion model (Equation 3), and (3) compute the observation update using the observation model (Equation 5), stopping after $T$ time steps. The resulting belief $bel(\boldsymbol{s}_T)$ represents the posterior probability distribution over likely goal locations.

# 4 Agent model

In this section, we describe our VLN agent that simultaneously: (1) builds a semantic spatial map from first-person views; (2) determines the most probable goal location in the current map by filtering likely trajectories taken by a human demonstrator from the start location (i.e., the 'ghost'); and (3) executes actions to reach the predicted goal. Each of these functions is the responsibility of a separate module which we refer to as the *mapper*, *filter*, and *policy*, respectively. We begin with the mapper.

## 4.1 Mapper

At each time step $t$, the mapper updates a learned semantic spatial map $\mathcal{M}_t \in \mathbb{R}^{M \times Y \times X}$ in the world coordinate frame from first-person views. This map is a grid-based metric representation in which each grid cell contains a $M$-sized latent vector representing the visual appearance of a small corresponding region in the environment. $X$ and $Y$ are the spatial dimensions of the semantic map, which could be dynamically resized if necessary. The map maintains a representation for every world coordinate $(x, y)$ that has been observed by the agent, and each map cell is computed from all past observations of the region. We define the world coordinate frame by placing the agent at the center of the map at the start of each episode, and defining the xy plane to coincide with the ground plane.

**Inputs.** As with previous work on VLN task [5–7], we provide the agent with a panoramic view of its environment at each time step[2] comprised of a set of RGB images $\mathcal{I}_t = \{I_{t,1}, I_{t,2}, \ldots, I_{t,K}\}$, where $I_{t,k}$ represents the image captured in direction $k$. The agent also receives the associated depth images $\mathcal{D}_t = \{D_{t,1}, D_{t,2}, \ldots, D_{t,K}\}$ and camera poses $\mathcal{P}_t = \{P_{t,1}, P_{t,2}, \ldots, P_{t,K}\}$. We additionally assume that the camera intrinsics and the ground plane are known. In the VLN task, these inputs are provided by the simulator, in other settings they could be provided by SLAM systems etc.

**Image processing.** Each image $I \in \mathbb{R}^{H \times W \times 3}$ is processed with a pretrained convolutional neural network (CNN) to extract a downsized visual feature representation $\boldsymbol{v} \in \mathbb{R}^{H' \times W' \times C}$. To extract a corresponding depth image $\boldsymbol{d} \in \mathbb{R}^{H' \times W'}$, we apply 2D adaptive average pooling to the original depth image $D \in \mathbb{R}^{H \times W}$. Missing (zero) depth values are excluded from the pooling operation.

**Feature projection.** Similarly to MapNet [12], we project CNN features $\boldsymbol{v}$ onto the ground plane in the world coordinate frame using the corresponding depth image $\boldsymbol{d}$, the camera pose $P$, and a pinhole camera model using known camera intrinsics. We then discretize the projected features into a 2D spatial grid $\mathcal{F}_t \in \mathbb{R}^{C \times Y \times X}$, using elementwise max pooling to handle feature collisions in a cell.

**Map update.** To integrate map observations $\mathcal{F}_t$ into our semantic spatial map $\mathcal{M}_t$, we use a convolutional implementation [31] of a Gated Recurrent Unit (GRU) [32]. In preliminary experiments we found that using convolutions in both the input-to-state and state-to-state transitions reduced the variance in the performance of the complete agent by sharing information across neighboring map cells. However, since both the map $\mathcal{M}_t$ and the map update $\mathcal{F}_t$ are sparse, we use a sparsity-aware convolution operation that evaluates only observed pixels and normalizes the output [33]. We also mask the GRU map update to prevent bias terms from accumulating in the unobserved regions.

## 4.2 Filter

At the beginning of each episode the agent is placed at a start location $s_0^* = (x_0, y_0, \theta_0)$, where $\theta$ represents the agent's heading and $x$ and $y$ are coordinates in the world frame as previously described. The agent is given an instruction $\mathcal{X}$ describing the trajectory to an unknown goal coordinate $s_T^* = (x_T, y_T, \cdot)$. As an intermediate step towards actually reaching the goal, we wish to identify likely goal locations in the partially-observed semantic spatial map $\mathcal{M}$ generated by the mapper.

Our approach to this problem is based on the observation that a natural language navigation instruction typically conveys a sequence of expected future observations and actions, as previously discussed. Based on this observation, we frame the problem of determining the goal location $s_T^*$ as a tracking problem. As illustrated in Figure 2 and described further below, we implement a Bayes filter to track the pose $s_t^*$ of a hypothetical human demonstrator (i.e., the 'ghost') from the start location to the goal. As inputs to the filter, we provided a series of latent observations $o_t$ and actions $a_t$ extracted from the navigation instruction $\mathcal{X}$. The output of the filter is the belief over likely goal locations $bel(s_T)$.

Note that in this section we use the subscript $t$ to denote time steps in the filter, overloading the notation from Section 4.1 in which $t$ referred to agent time steps. We wish to make clear that in our model the filter runs in an inner loop, re-estimating belief over trajectories taken by a demonstrator starting from $s_0$ each time the map is updated by the agent in the outer loop.

**Belief.** We define the state $s_t = (x_t, y_t, \theta_t)$ using the agent's $(x, y)$ position and heading $\theta$. We represent the belief over the demonstrator's state at each time step $t$ with a histogram, implemented as a tensor $bel(s_t) = b_t$, $b_t \in \mathbb{R}^{\Theta \times Y \times X}$ where $X$, $Y$ and $\Theta$ are the number of bins for each component of the state, respectively. Using a histogram-based approach allows the filter to track multiple hypotheses, meshes easily with our implementation of a grid-based semantic map, and leads naturally to an efficient motion model implementation based on convolutions, as discussed further below. However, our proposed approach could also be implemented as a particle filter [29, 30], for example if discretization error was a significant concern.

**Observations and actions.** To transform the instruction $\mathcal{X}$ into a latent representation of observations $o$ and actions $a$, we use a sequence-to-sequence model with attention [34]. We first tokenize the instruction into a sequence of words $\mathcal{X} = \{x_1, x_2, \ldots, x_l\}$ which are encoded using learned word embeddings and a bi-directional LSTM [19] to output a series of encoder hidden states $\{e_1, e_2, \ldots, e_l\}$ and a final hidden state $e$ representing the output of a complete pass in each direction. We then use an LSTM decoder to generate a series of latent observation and action vectors $\{o_1, o_2, \ldots, o_T\}$ and $\{a_1, a_2, \ldots, a_T\}$ respectively. Here, $o_t$ is given $o_t = [\hat{e}_t^o, h_t]$, where $h_t$ is the hidden state of the decoder LSTM, and $\hat{e}_t^o$ is the attended instruction representation computed using a standard dot-product attention mechanism [35]. The action vectors $a_t$ are computed analogously, using the same decoder LSTM but with a separate learned attention mechanism. The only input to the decoder LSTM is a positional encoding [36] of the decoding time step $t$. While the correct number of decoding time steps $T$ is unknown, in practice we always run the filter for a fixed number of time steps equal to the maximum trajectory length in the dataset (which is 6 steps in the navigation graph).

**Motion model.** We implement the *motion model* $p(s_t \mid s_{t-1}, a_t, \mathcal{M})$ as a convolution over the belief $b_{t-1}$. This ensures that agent motion is consistent across the state space while explicitly enforcing locality, i.e., the agent cannot move further than half the kernel size in a single time step. Similarly to Jonschkowski and Brock [15], the prediction step from Equation 1 is thus reformulated as:

$$\overline{b}_t = b_{t-1} * g(a_t, \mathcal{M}) \tag{3}$$

where we define an action- and map-dependent motion kernel $g(a_t, \mathcal{M}) \in \mathbb{R}^{\Theta^2 \times M^2}$ given by:

$$g(a_t, \mathcal{M}) = \text{softmax}(\text{conv}([a_t, \mathcal{M}])) \tag{4}$$

where conv is a small 3-layer CNN with ReLU activations operating on the semantic spatial map $\mathcal{M}$ and the spatially-tiled action vector $a_t$, $M$ is the motion kernel size and the softmax function enforces the prior that $g(a_t, \mathcal{M})$ represents a probability mass function. Note that we include $\mathcal{M}$ in the input so that the motion model can learn that the agent is unlikely to move through obstacles.

**Observation model.** We require an *observation model* $p(o_t \mid s_t, \mathcal{M})$ to define the likelihood of a latent observation $o_t$ conditioned on the agent's state $s_t$ and the map $\mathcal{M}$. A generative observation model like this would be hard to learn, since it is not clear how to generate high-dimensional latent observations and normalization needs to be done across observations, not states. Therefore, we follow

prior work [30] and learn a discriminative observation model that takes $\boldsymbol{o}_t$ and $\mathcal{M}$ as inputs and directly outputs the likelihood of this observation for each state. As detailed further in Section 4.4, this observation model is trained end-to-end without direct supervision of the likelihood.

To implement our observation model we use LingUNet [2], a language-conditioned image-to-image network based on U-Net [37]. Specifically, we use the LingUNet implementation from Blukis et al. [11] with 3 cascaded convolution and deconvolution operations. The spatial dimensionality of the LingUNet output matches the input image (in this case, $\mathcal{M}$), and number of output channels is selected to match the number of heading bins $\Theta$. Outputs are restricted to the range $[0, 1]$ using a sigmoid function. The observation update from Equation 2 is re-defined as:

$$\boldsymbol{b}_t = \eta \, \overline{\boldsymbol{b}}_t \odot \text{LingUNet}(\boldsymbol{o}_t, \mathcal{M}) \tag{5}$$

where $\eta$ is a normalization constant and $\odot$ represents element-wise multiplication.

**Goal prediction.** In summary, to identify goal locations in the partially-observed spatial map $\mathcal{M}$, we initialize the belief $\boldsymbol{b}_0$ with the known starting state $\boldsymbol{s}_0$. We then iteratively: (1) Generate a latent observation $\boldsymbol{o}_t$ and action $\boldsymbol{a}_t$, (2) Compute the prediction step using Equation 3, and (3) Compute the observation update using Equation 5. We stop after $T$ filter update time steps. The resulting belief $\boldsymbol{b}_T$ represents the posterior probability distribution over goal locations.

### 4.3 Policy
The final component of our agent is a simple reactive policy network. It operates over a global action space defined by the complete set of panoramic viewpoints observed in the current episode (including both visited viewpoints, and their immediate neighbors). Our agent thus memorizes the local structure of the observed navigation graph to enable it to return to any previously observed location in a single action. The probability distribution over actions is defined by a softmax function, where the logit associated with each viewpoint $i$ is given by $y_i = \text{MLP}([\boldsymbol{b}_{1:T,i}, \boldsymbol{v}_i])$, where MLP is a two-layer neural network, $\boldsymbol{b}_{1:T,i}$ is a vector containing the belief at each time step $1 : T$ in a gaussian neighborhood around viewpoint $i$, and $\boldsymbol{v}_i$ is a vector containing the distance from the agent's current location to viewpoint $i$, and an indicator variable for whether $i$ has been previously visited. If the policy chooses to revisit a previously visited viewpoint, we interpret this as a `stop` action. Note that our policy does not have direct access to any representation of the instruction, or the semantic map $\mathcal{M}$. Although our policy network is specific to the Matterport3D simulator environment, the rest of our pipeline is general and operates without knowledge of the simulator's navigation graph.

### 4.4 Learning
Our entire agent model is fully differentiable, from policy actions back to image pixels via the semantic spatial map, geometric feature projection function, etc. Training data for the model consists of instruction-trajectory pairs $(\mathcal{X}, \boldsymbol{s}^*_{1:T})$. In all experiments we train the filter using supervised learning by minimizing the KL-divergence between the predicted belief $\boldsymbol{b}_{1:T}$ and the true trajectory from the start to the goal $\boldsymbol{s}^*_{1:T}$, backpropagating gradients through the previous belief $\boldsymbol{b}_{t-1}$ at each step. Note that the predicted belief $\boldsymbol{b}_{1:T}$ is independent of the agent's actual trajectory $\boldsymbol{s}_{1:T}$ given the map $\mathcal{M}$. In the goal prediction experiments (Section 5.2), the model is trained without a policy and so the agent's trajectory $\boldsymbol{s}_{1:T}$ is generated by moving towards the goal with 50% probability, or randomly otherwise. In the full VLN experiments (Section 5.3), we train the filter concurrently with the policy. The policy is trained with cross-entropy loss to maximize the likelihood of the ground-truth target action, defined as the first action in the shortest path from the agent's current location $\boldsymbol{s}_t$ to the goal $\boldsymbol{s}^*_T$. In this regime, trajectories are generated by sampling an action from the policy with 50% probability, or selecting the ground-truth target action otherwise. In both sets of experiments we train all parameters end-to-end (except for the pretrained CNN). We have verified that the stand-alone performance of the filter is not unduly impacted by the addition of the policy, but we leave the investigation of more sophisticated RL training regimes to future work.

**Implementation details.** We provide further implementation details in the supplementary material. PyTorch code will be released to replicate all experiments.[3]

## 5 Experiments

### 5.1 Environment and dataset
**Simulator.** We use the Matterport3D Simulator [1] based on the Matterport3D dataset [17] containing RGB-D images, textured 3D meshes and other annotations captured from 11K panoramic viewpoints

Table 1: Goal prediction results given a natural language navigation instruction and a fixed trajectory that either moves towards the goal, or randomly, with 50:50 probability. We evaluate predictions at each time step, although on average the goal is not seen until later time steps. Our filtering approach that explicitly models trajectories outperforms LingUNet [2, 11] across all time steps (i.e., regardless of map sparsity). We confirm that add heading $\theta$ to the filter state provides a robust boost.

| | Val-Seen | | | | | | | | | Val-Unseen | | | | | | | | |
|---|---|---|---|---|---|---|---|---|---|---|---|---|---|---|---|---|---|---|
| Time step | 0 | 1 | 2 | 3 | 4 | 5 | 6 | 7 | Avg | 0 | 1 | 2 | 3 | 4 | 5 | 6 | 7 | Avg |
| Map Seen (m$^2$) | 47.2 | 62.5 | 73.3 | 82.1 | 90.7 | 98.3 | 105 | 112 | 83.9 | 45.6 | 60.3 | 69.8 | 78.0 | 84.9 | 91.1 | 96.7 | 102 | 78.6 |
| Goal Seen (%) | 8.82 | 17.2 | 25.9 | 33.7 | 41.2 | 48.8 | 54.5 | 60.2 | 36.3 | 16.0 | 25.2 | 34.6 | 43.2 | 50.5 | 57.0 | 62.8 | 67.6 | 44.6 |
| **Prediction Error (m)** | | | | | | | | | | | | | | | | | | |
| Hand-coded baseline | 7.42 | 7.33 | 7.19 | 7.18 | 7.15 | 7.13 | 7.09 | 7.11 | 7.20 | 6.75 | 6.53 | 6.40 | 6.37 | 6.29 | 6.20 | 6.15 | 6.12 | 6.35 |
| LingUNet baseline | 7.17 | 6.66 | 6.17 | 5.75 | 5.42 | 5.15 | 4.89 | 4.69 | 5.74 | 6.18 | 5.80 | 5.40 | 5.17 | 4.90 | 4.65 | 4.44 | 4.27 | 5.10 |
| Filter, $s = (x, y)$ (ours) | 6.45 | 5.94 | 5.66 | 5.25 | 5.00 | 4.86 | 4.67 | 4.62 | 5.31 | 5.92 | 5.50 | 5.14 | 4.88 | 4.67 | 4.45 | 4.41 | 4.30 | 4.91 |
| Filter, $s = (x, y, \theta)$ (ours) | **6.10** | **5.75** | **5.30** | **5.06** | **4.81** | **4.71** | **4.59** | **4.46** | **5.09** | **5.69** | **5.28** | **4.90** | **4.60** | **4.40** | **4.26** | **4.14** | **4.05** | **4.67** |
| **Success Rate (<3m error)** | | | | | | | | | | | | | | | | | | |
| Hand-coded baseline | 17.3 | 17.8 | 18.5 | 18.2 | 18.0 | 19.1 | 18.8 | 18.6 | 18.3 | 18.9 | 20.1 | 21.1 | 21.3 | 21.8 | 22.2 | 22.6 | 22.9 | 21.4 |
| LingUNet baseline | 10.7 | 16.7 | 21.2 | 25.8 | 29.7 | 33.6 | 36.9 | 39.1 | 26.7 | 16.9 | 22.3 | 27.7 | 31.6 | 35.2 | 38.4 | 41.1 | 44.5 | 32.2 |
| Filter, $s = (x, y)$ (ours) | 24.6 | 29.3 | 31.9 | 35.9 | 39.7 | 41.0 | 42.1 | 41.2 | 35.7 | 29.1 | 32.5 | 36.1 | 39.2 | 41.9 | 44.5 | 45.7 | 46.2 | 39.4 |
| Filter, $s = (x, y, \theta)$ (ours) | **30.9** | **34.3** | **38.4** | **41.6** | **43.7** | **44.9** | **44.3** | **46.2** | **40.6** | **34.2** | **38.7** | **42.7** | **46.1** | **48.2** | **48.4** | **49.9** | **51.2** | **44.9** |

densely sampled throughout 90 buildings. Using this dataset, the simulator implements a visually-realistic first-person environment that allows the agent to look in any direction while moving between panoramic viewpoints along edges in a navigation graph. Viewpoints are 2.25m apart on average.

**Depth outputs.** As the Matterport3D Simulator supports RGB output only, we extend it to support depth outputs which are necessary to accurately project CNN features into the semantic spatial map. Our simulator extension projects the undistorted depth images from the Matterport3D dataset onto cubes aligned with the provided 'skybox' images, such that each cube-mapped pixel represents the euclidean distance from the camera center. We then adapt the existing rendering pipeline to render depth images from these cube-maps, converting depth values from euclidean distance back to distance from the camera plane in the process. To fill missing depth values corresponding to shiny, bright, transparent, and distant surfaces, we apply a simple cross-bilateral filter based on the NYUv2 implementation [38]. We additionally implement various other performance improvements, such as caching, which boosts the frame-rate of the simulator up to 1000 FPS, subject to GPU performance and CPU-GPU memory bandwith. We have incorporated these extensions into the original simulator codebase.[4]

**R2R instruction dataset.** We evaluate using the Room-to-Room (R2R) dataset for Vision-and-Language Navigation (VLN) [1]. The dataset consists of 22K open-vocabulary, crowd-sourced navigation instructions with an average length of 29 words. Each instruction corresponds to a 5–24m trajectory in the Matterport3D dataset, traversing 5–7 viewpoint transitions. Instructions are divided into splits for training, validation and testing. The validation set is further split into two components: val-seen, where instructions and trajectories are situated in environments seen during training, and val-unseen containing instructions situated in environments that are not seen during training. All the test set instructions and trajectories are from environments that are unseen in training and validation.

## 5.2  Goal prediction results
We first evaluate the goal prediction performance of our proposed mapper and filter architecture in a policy-free setting using fixed trajectories. Trajectories are generated by an agent that moves towards the goal with 50% probability, or randomly otherwise. As an ablation, we also report results for our model excluding heading from the agent's filter state, i.e., $s_t = (x, y)$, to quantify the value of encoding the agent's orientation in the motion and observation models. We compare to two baselines as follows:

**LingUNet baseline.** As a strong neural net baseline, we compare to LingUNet [2] – a language-conditioned variant of the U-Net image-to-image architecture [37] – that has recently been applied to goal location prediction in the context of a simulated quadrocopter instruction-following task [11]. We choose LingUNet because existing VLN models [3–9] do not explicitly model the goal location or the map, and are thus not capable of predicting the goal location from a provided trajectory. Following Blukis et al. [11] we train a 5-layer LingUNet module conditioned on the sentence encoding $e$ and

Table 2: Results for the full VLN task on the R2R dataset. Our model achieves credible results for a new model class trained exclusively with imitation learning (no `RL`) and without any data augmentation or specialized pretraining (`Aug`).

| Model | RL | Aug | Val-Seen | | | | | Val-Unseen | | | | | Test | | | | |
|---|---|---|---|---|---|---|---|---|---|---|---|---|---|---|---|---|---|
| | | | TL | NE | OS | SR | SPL | TL | NE | OS | SR | SPL | TL | NE | OS | SR | SPL |
| RPA [4] | ✓ | | 8.46 | 5.56 | 0.53 | 0.43 | - | 7.22 | 7.65 | 0.32 | 0.25 | - | 9.15 | 7.53 | 0.32 | 0.25 | 0.23 |
| Speaker-Follower [7] | | ✓ | - | 3.36 | 0.74 | 0.66 | - | - | 6.62 | 0.45 | 0.36 | - | 14.82 | 6.62 | 0.44 | 0.35 | 0.28 |
| RCM [3] | ✓ | | 10.65 | 3.53 | 0.75 | 0.67 | - | 11.46 | 6.09 | 0.50 | 0.43 | - | 11.97 | 6.12 | 0.50 | 0.43 | 0.38 |
| Self-Monitoring [5] | | ✓ | - | 3.18 | 0.77 | 0.68 | 0.58 | - | 5.41 | 0.59 | 0.47 | 0.34 | 18.04 | 5.67 | 0.59 | 0.48 | 0.35 |
| Regretful Agent [6] | | ✓ | - | 3.23 | 0.77 | 0.69 | 0.63 | - | 5.32 | 0.59 | 0.50 | 0.41 | 13.69 | 5.69 | 0.56 | 0.48 | 0.40 |
| FAST [9] | ✓ | | - | - | - | - | - | 21.1 | 4.97 | - | 0.56 | 0.43 | 22.08 | **5.14** | **0.64** | **0.54** | 0.41 |
| Back Translation [8] | ✓ | ✓ | 11.0 | 3.99 | - | 0.62 | 0.59 | 10.7 | 5.22 | - | 0.52 | 0.48 | 11.66 | 5.23 | 0.59 | 0.51 | **0.47** |
| Speaker-Follower [7] | | | - | 4.86 | 0.63 | 0.52 | - | - | 7.07 | 0.41 | 0.31 | - | - | - | - | - | - |
| Back Translation [8] | | | 10.3 | 5.39 | - | 0.48 | 0.46 | 9.15 | 6.25 | - | 0.44 | 0.40 | - | - | - | - | - |
| Ours | | | 10.15 | 7.59 | 0.42 | 0.34 | 0.30 | 9.64 | 7.20 | 0.44 | 0.35 | 0.31 | 10.03 | 7.83 | 0.42 | 0.33 | 0.30 |

the semantic map $\mathcal{M}$ to directly predict the goal location distribution (as well as a path visitation distribution, as an auxilliary loss) in a single forward pass. As we implement our observation model using a (smaller, 3-layer) LingUNet, the LingUNet baseline resembles an ablated single-step version of our model that dispenses with the decoder generating latent observations and actions as well as the motion model. Note that we use the same mapper architecture for our filter and for LingUNet.

**Hand-coded baseline.** We additionally compare to hand-coded goal prediction baseline designed to exploit biases in the R2R dataset [1] and the provided trajectories. We first calculate the mean straight-line distance from the start position to the goal across the entire training set, which is 7.6m. We then select as the predicted goal the position $(x, y)$ in the map at a radius of 7.6m from the start position that has the greatest observed map area in an Gaussian-weighted neighborhood of $(x, y)$.

**Results.** As illustrated in Table 1, our proposed filter architecture that explicitly models belief over trajectories that could be taken by a human demonstrator outperforms a strong LingUNet baseline at predicting the goal location (with an average success rate of 45% vs. 32% in unseen environments). This finding holds at all time steps (i.e., regardless of the sparsity of the map). We also demonstrate that removing the heading $\theta$ from the agent's state in our model degrades this success rate to 39%, demonstrating the importance of relative orientation to instruction understanding. For instance, it is unlikely for an agent following the true path to turn 180 degrees midway through (unless this is commanded by the instruction). Similarly, without knowing heading, the model can represent instructions such as 'go past the table' but not 'go past with the table on your left'. Finally, the poor performance of the handcoded baseline confirms that the goal location cannot be trivially predicted from the trajectory.

## 5.3 Vision-and-Language Navigation results

Having established the efficacy of our approach for goal prediction from a partial map, we turn to the full VLN task that requires our agent to take actions to actually reach the goal.

**Evaluation.** In VLN, an episode is successful if the final navigation error is less than 3m. We report our agent's average success rate at reaching the goal (`SR`), and `SPL` [16], a recently proposed summary measure of an agent's navigation performance that balances navigation success against trajectory efficiency (higher is better). We also report trajectory length (`TL`) and navigation error (`NE`) in meters, as well as oracle success (`OS`), defined as the agent's success rate under an oracle stopping rule.

**Results.** In Table 2, we present our results in the context of state-of-the-art methods; however, as noted by the `RL` and `Aug` columns in the table, these approaches include reinforcement learning and complex data augmentation and pretraining strategies. These are non-trivial extensions that are the result of a community effort [3–9] and are orthogonal to our own contribution. We also use a less powerful CNN (ResNet-34 vs. ResNet-152 in prior work). For the most direct comparison, we consider the ablated models in the lower panel of Table 2 to be most appropriate. We find these results promising given this is the first work to explore such a drastically different model class (i.e., maintaining a metric map and a probability distribution over alternative trajectories in the map). Our model also exhibits less overfitting than other approaches – performing equally well on both seen (val-seen) and unseen (val-unseen) environments.

Further, our filtering approach allows us greater insight into the model. We examine a qualitative example in Figure 3. On the left, we can see the agent attends to appropriate visual and direction words when generating latent observations and actions, supporting the intuition in Figure 1. On the

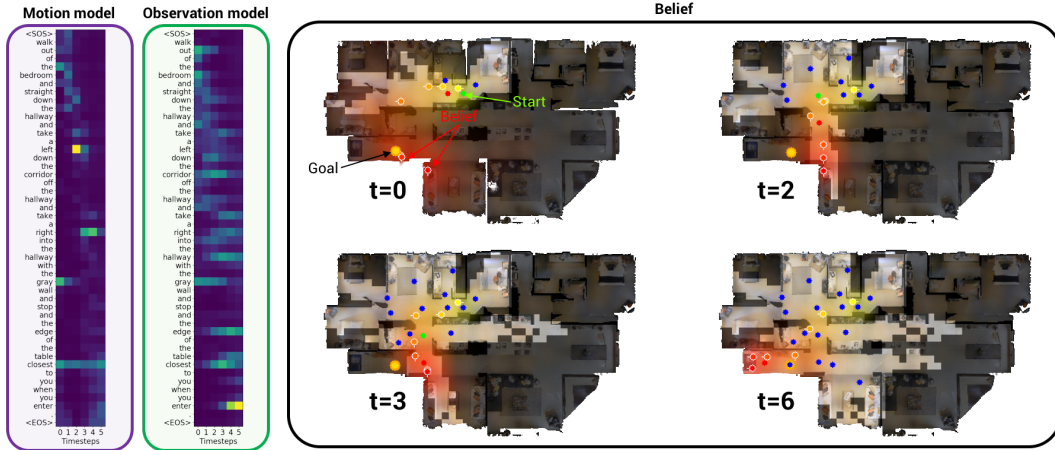

Figure 3: Left: Textual attention during latent observation and action generation is appropriately more focused towards action words ('left', 'right') for the motion model, and visual words ('bedroom', 'corridor', 'table') for the observation model. Right: Top-down view illustrating the agent's expanding semantic spatial map (lighter-colored region), navigation graph (blue dots) and corresponding belief (red heatmap and circles with white heading markers) when following this instruction. At $t = 0$ the map is largely unexplored, and the belief is approximately correct but dispersed. By $t = 6$, the agent has become confident about the correct goal location, despite many now-visible alternative paths.

right, we can see the growing confidence our goal predictor places on the correct location as more of the map is explored – despite the increasing number of visible alternatives. We provide further examples (including insight into the motion and observation models) in the supplementary video.

## 6   Conclusion

We show that instruction following can be formulated as Bayesian state tracking in a model that maintains a semantic spatial map of the environment, and an explicit probability distribution over alternative possible trajectories in that map. To evaluate our approach we choose the complex problem of Vision-and-Language Navigation (VLN). This represents a significant departure from existing work in the area, and required augmenting the Matterport3D simulator with depth. Empirically, we show that our approach outperforms recent alternative approaches to goal location prediction, and achieves credible results on the full VLN task without using RL or data augmentation – while offering reduced overfitting to seen environments, unprecedented intepretability and less reliance on the simulator's navigation constraints.

### Acknowledgments

We thank Abhishek Kadian and Prithviraj Ammanabrolu for their help in the initial stages of the project. The Georgia Tech effort was supported in part by NSF, AFRL, DARPA, ONR YIPs, ARO PECASE. The views and conclusions contained herein are those of the authors and should not be interpreted as necessarily representing the official policies or endorsements, either expressed or implied, of the U.S. Government, or any sponsor.

## Footnotes

*First two authors contributed equally.

[2]The panoramic setting is chosen for comparison with prior work – not as a requirement of our architecture.

[3]https://github.com/batra-mlp-lab/vln-chasing-ghosts

[4]https://github.com/peteanderson80/Matterport3DSimulator

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
