[Supplementary Material · supplementary.pdf]



# Chasing Ghosts: Instruction Following as Bayesian State Tracking

## Supplementary Material

## Implementation Details

**Simulator.** In experiments, we set the Matterport3D simulator [1] to generate $320 \times 256$ pixel images with a 60 degree vertical field of view. To capture more of the floor and nearby obstacles (and less of the roof) we set the camera elevation to 30 degrees down from horizontal. At each panoramic viewpoint location in the simulator we capture a horizontal sweep containing 12 images at 30 degree increments, which are projected into the map in a single time step as described in Section 4.1 of the main paper.

**Mapper.** For our CNN implementation we use a ResNet-34 [2] architecture that is pretrained on ImageNet [3]. We found that fine-tuning the CNN while training our model mainly improved performance on the Val-Seen set, and so we left the CNN parameters fixed in the reported experiments. To extract the visual feature representation $v$ we concatenate the output from the CNN's last 2 layers to provide a $16 \times 20 \times 768$ representation. The dimensionality of our map representation $\mathcal{M}$ is fixed at $128 \times 96 \times 96$ and each cell represents a square region with side length $0.5$m (the entire map is thus 48m $\times$ 48m). In the mapper's convolutional [4] GRU [5] we use $3 \times 3$ convolutional filters and we train with spatial dropout [6] of $0.5$ in both the input-to-state and state-to-state transitions with fixed dropout masks for the duration of each episode.

**Filter.** In the instruction encoder we use a hidden state size of 256 for both the forward and backward encoders, and a word embedding size of 300. We use a motion kernel size $M$ of 7, but we upscale the motion kernel $g(\boldsymbol{a}_t, \mathcal{M})$ by a scale factor of $2\times$ before applying it such that the agent can move a maximum of 3.5m in a single time step.

**Training.** In training, we use the Adam optimizer [7] with an initial learning rate of 1e-3, weight decay of 1e-7, and a batch size of 5. In the goal prediction experiment, all models are trained for 8K iterations, after which all models have converged. In the full VLN experiment, our models are trained for 17.5K iterations, and we pick the iteration with the highest SPL performance on Val-Unseen to report and submit to the test server. Training the model takes around 1 day for goal prediction, and 2.5 days for the full VLN task, using a single Titan X GPU.

**Visualizations.** In the main paper and the supplementary video, we depict top-down floorplan visualizations of Matterport environments to provide greater insight into the model's behavior. These visualizations are rendered from textured meshes in the Matterport3D dataset [8], using the provided GAPS software which was modified to render using an orthographic projection.

## Visualizations

**Observations and actions.** In this section we provide further visualizations of the attention weights in the sequence decoders that generate latent observations and actions (refer to Section 4.2 of the main paper). Instructions are examples from the Val-Unseen set. In general, the attention models for the motion model (generating latent action vectors $\boldsymbol{a}$) and the observation model (generating latent observation vectors $\boldsymbol{o}$) specialize in different ways. The motion model focuses attention on action words, while the observation model focuses on visual words, as illustrated in Figures 1 and 2. The sequential ordering of the instructions (e.g., attention weights showing a diagonal structure from top-left to bottom-right) is also evident.

Figure 1: Example attention weight visualizations from the motion model and observation model inputs. The motion model focuses on action descriptions, such as 'go through' (top left) and 'turn right' (bottom left and right). In contrast, the observation model focuses more towards visual words such as 'kitchen' (top left), 'toilet' (top right), and 'hallway' (bottom left and right).

Figure 2: More example attention weight visualizations from the motion and observation models. Here, the motion model focuses on action descriptions, such as 'left', while the observation model focuses attention on visual words such as 'couch' (bottom left), 'table and chairs' (bottom right), and 'door' (bottom right).