[Reviews · NeurIPS 2019]

Reviewer 1



This paper presents a new model for the task of vision-and-language navigation (VLN), where an agent must follow a natural language navigation instruction in a simulated environment with real-world visual imagery. The model combines several components: (1) a learned, spatial memory representation of the environment, (2) a neuralized Bayes' filter for goal localization that conditions on latent "actions" and "observations", which are produced by a recurrent decoder that conditions on the instruction, and (3) a reactive policy that conditions on the goal predicted by the Bayes' filter. The paper trains the model end-to-end and evaluates it on the benchmark Room-to-Room dataset, in both the standard navigation setting and in goal prediction. *Originality* While, as the paper acknowledges, all the individual components combined here have been explored in some form in past work (the metric spatial memory, differentiable Bayes filter, and rollouts of latent observations and actions), the combination and application to the language-conditioned navigation task is, to my knowledge, novel. I found it creative and well-motivated. *Quality* The main weakness of the paper is the results on the full navigation task, which are weak in comparison to past work. As the paper points out, this past work has used a variety of training and inference conditions which improve performance, and are likely orthogonal to the contributions here. However, much of this past work has also reported results without these augmentations, and those results are comparable or better than the navigation performance here. It would be clearer if the paper presented these results (for example the "co-grounding" and "greedy decoding" ablation of Ma et al. which obtains 42 SR and 28 SPL on the val-unseen environments, and the Behavioral Cloning (IL) ablation of Tan et al, which obtains 43.6 SR and 40 SPL on val-unseen) rather than the augmented settings, or explained why they are not comparable. In particular, since this paper uses the panoramic state representation of Fried et al, and an action space similar to theirs, it seems that their "panoramic space" ablation model might be a more appropriate baseline than the non-panoramic Seq2Seq model compared to here. However, all these differences seem at least partly explainable due to the use of different ResNet visual features than these past works. In addition, the results on the goal prediction task show a substantial improvement over the strong LingUNet model. *Clarity* I found the paper overall extremely clear about the model details and the intuition for each part, and the motivation for the work. There were a few minor details about the training procedure that were underspecified: - Is the true state sequence in 245 always the human trajectory, or does it include the exploration that is done by the model during training? - When training the policy with cross-entropy loss, are the parameters of the rest of the network (e.g. the filter and semantic map) also updated (or are they just updated by the filter supervision)? - Is the mapper-filter in the experiments 5.2 produced by training without the policy, or does this take a trained full model and remove the policy component? (it seems likely to be the first one, but the text is a bit unclear) *Significance* While the results on the full navigation task don't show an improvement over past work, I think that the model class is still likely to be built upon by researchers in this area. Past work has seen two high level problems in this task, which models like this one may be able to address: (1) Substantial improvements from exploration of the environment during inference time. Having a model with an explicit simulated planning component makes it possible to see how much the simulated planning using the learned environment representation could reduce the need for actual exploration. (2) A generalization gap in novel environments. It seems promising that this model has no gap in performance between seen and unseen environments, although the reason for this is not explained in this work. *Minor comments* - 238: The policy does seem to indirectly have access to a representation of the instruction and semantic map through the belief network; this could be clarified by saying that it has no direct access. - I found it surprising that incorporating the agent's heading into the belief state has such a large impact on performance, given the panoramic visual representation and action space. Some discussion of this would be helpful.

Reviewer 2



Summary: In this work, the authors propose a modular pipeline, reminiscent of traditional control architectures in robotics, in which a map is updated online, a state-belief distribution is maintained over the map through sequential filtering, and a policy is conditioned on this belief. This architecture is demonstrated on a vision-and-language navigation task, although in principle could apply to other spatial tasks as well. Originality: Neither map->estimate->control pipelines nor end-to-end differentiable Bayesian filtering are new, as the authors note, but this application is a novel and promising avenue for this sort of task. In particular, conditioning a reactive policy on the Bayesian state estimate has the potential to avoid overfitting, since its input has been abstracted away from the observations. The performance results in the paper are poor in the absolute sense, but do seem to show less overfitting. The action space of the agent is also novel, although it requires significant domain and task knowledge in the form of the structure of the navigation graph. Quality: The technical contribution is sound. The results fall significantly below state-of-the-art algorithms. Perhaps performance could be increased by techniques such as reinforcement learning and data augmentation, as in the state-of-the-art approaches, although this is not clear. Clarity: The paper is well-written and organized. The video attachment is helpful for understanding. Significance: With such low performance results, it is not clear that this work directly advances the field in this area, despite its novelty.

Reviewer 3



The authors propose a new method for inferring vision and language navigation using Bayesian filtering and state tracking . The paper was well written up to some equation nomenclature and acronyms but still very easy to follow and I think it is indeed innovative to extend the simulator for the purpose of learning as a synthesis of two different fields. This submission would definitely advance the fields, the ability to generate more realistic training grounds for VLN model can present a huge advantage. The experiments were also truly convincing (nice touch on the videos of the problem). Yet there are a few things I would be happy to have been clarified or that are unclear. The first and I think by biggest question is how does the state rejection mechanism compare to the one of Weib and et el https://arxiv.org/abs/1511.06458. Can one think of the language component as a refined prior ? In the abstract can you please expand on what do you mean by a strong baseline? Line 134 what are XY ? Line 147-150 is a bit of a mess of definitions and notation

[Author Response · NeurIPS 2019]

We thank the reviewers for their thoughtful comments! We are encouraged that all reviewers voted to accept, finding
the paper to be creative and well-motivated [**R1**], extremely clear / well-written [**R1 R2 R3**], and that the technical
contributions of the paper – a novel model for language-conditioned instruction following – was judged to have
medium-high [**R1**] to high [**R2**] significance. We address specific concerns below (and will incorporate all feedback).

[**R1**] **Comparison to prior work.** Thanks for pointing out the relevant ablated comparisons in [4] and [7], which
achieve val-unseen success rates (SR) of 42% and 43.6% respectively. With additional hyperparameter tuning, we found
the results of our model increased to 33% SR / 30% SPL on val-unseen (up from 31% / 27%). Compared to these works,
we use less powerful ResNet-34 image features (instead of ResNet-152) and our model is much less dependent on the
structure of the navigation graph (see L36-41 below). We will update Table 2 with these comparisons instead of [1].

[**R1**] **Impact of agent heading.** Excellent point for which we will add discussion. Heading indicates important cues
about a trajectory which the model can leverage in the belief state. For instance, is is unlikely for an agent following the
true path to turn 180 degrees midway through (unless this is commanded by the instruction). Similarly, without knowing
heading, the model can represent instructions such as 'go past the table' but not 'go past with the table on your left'.

[**R1**] **Are the mapper-filter experiments in Sec 5.2 produced by training without the policy? Are the filter and**
**semantic map parameters updated while training with the policy in Sec 5.3?** Yes, to both. In Sec 5.2 we train
without the policy, using only the filter loss (KL-divergence between the predicted belief $b_{1:T}$ and the true state $s^*_{1:T}$). In
Sec 5.3 we train concurrently with both the filter loss and the policy loss (cross-entropy loss to maximize the likelihood
of the ground-truth target action). In both cases we train all parameters end-to-end (except for the pretrained CNN). We
have verified that the stand-alone performance of the filter is not unduly impacted by the addition of the policy.

[**R1**] **Is the true state sequence $s^*_{1:T}$ (L245) always the human trajectory, or does it include the exploration that**
**is done by the agent during training?** Yes it is always the ground-truth (human) trajectory. Recall that the filter
subcomponent is simply trying to estimate the path of an ideal agent following the instructions, given a partially-
observed semantic map $\mathcal{M}$. However, the agent's exploration does determine the content of the map (i.e., the map
contains the observations from the agent's explorations in the current episode). We will clarify.

[**R1**] **No 'direct' access to the instruction and map** We will add this qualification at L238.

[**R2**] **"Middling" performance compared to SoTA.** As in the paper, we freely admit that the proposed method does
not approach SoTA performance, but we agree with the reviewer guidelines in that *"Solid, technical papers that explore*
*new territory or point out new directions for research are preferable to papers that advance the state of the art, but*
*only incrementally."* Moreover, our novel formulation for instruction-conditioned goal-identification in VLN as state
tracking provides increased inspectability of agent beliefs and reasons over 2D space rather than just navigable nodes.
As such, it may be of broader interest outside the VLN task.

[**R2**] **Lack indications that this architecture will lead to SoTA.** The performance of the model improved with
additional hyperparameter tuning (see L5-9 above). Given the rebuttal period, it was impractical to incorporate the
orthogonal additions like RL training, data augmentation from a trained speaker model, or trajectory re-ranking which
have lead to much of the advancement at VLN.

[**R2**] **The action space of the agent... requires significant domain and task knowledge in the form of the structure**
**of the navigation graph.** We would like to clarify that the navigation graph is defined by the R2R dataset [1]. Previous
works [2-8] use LSTMs that operate directly over the nodes in this graph. Our approach is far less reliant on this
structure than prior work (at test time, our mapper and filter operate on arbitrarily discretized space, with no knowledge
of the navigation graph). We anticipate this could be an advantage for sim-to-real transfer (i.e., in real robot scenarios
where a navigation graph is not provided, and could be non-trivial to generate).

[**R3**] **How does the state rejection mechanism compare to the one of Weib et el. arxiv.org/abs/1511.06458. Can**
**one think of the language component as a refined prior?** We are not sure what the reviewer is referring by a "state
rejection mechanism" in our model, but are happy to hear more details in an updated review. With regard to Weib et al. -
we do not need to employ such rejection or importance sampling techniques because we use a histogram representation
for the posterior, not a set of weighted samples as in particle-based Bayesian state tracking. At a high-level, we agree
that language does define the set of expected observation and actions an agent should take throughout the trajectory.

[**R3**] **At L134 what are XY?** $X$ and $Y$ are the initial spatial dimensions of the semantic map in grid cells. In
experiments this dimension is 96 (refer supplementary). In practice the map could be dynamically resized if necessary.

[**R3**] **What does 'strong baseline' in abstract refer to?** We are referring to the LingUNet model and the Hand-coded
baseline that exploits dataset biases (Sec 5.2). Note that existing models [2-8] do not explicitly model the goal location,
and are thus not capable of predicting the goal location from a provided sequence of observations.

[Meta-Review · NeurIPS 2019]

Interesting new approach for instruction-based navigation using Bayesian state tracking.